# Uncertainty-Routed Human–LLM Curation and Calibration for ANLI

## Abstract

Adversarial NLI (ANLI) reveals distribution-shift failures that static benchmarks miss, motivating evaluation and curation that are explicitly uncertainty-aware. We present $\text{URC}^2$—**Uncertainty-Routed Curation & Calibration**, a three-stage pipeline that improves dataset quality and model reliability. $\text{URC}^2$ decomposes per-example predictive uncertainty into *aleatoric* (data/label ambiguity) and *epistemic* (model uncertainty, measured via mutual information) using a three-teacher ensemble (DeBERTa-v3-large, RoBERTa-large, XLM-R-large). A two-lane relabeling workflow then routes cases: a *Human lane* relabels, removes, or down-weights aleatoric-heavy examples, while an *LLM lane* adjudicates epistemic-heavy examples using instruction-tuned self-consistency checks. Curated labels and per-example weights drive a lightweight retraining and recalibration loop for each teacher, yielding an updated ensemble. On ANLI, $\text{URC}^2$ reduces development-set expected calibration error by **30%** (to **0.146**) and lowers corpus-level uncertainty *without* degrading accuracy. Unlike prior curation pipelines that treat uncertainty monolithically, $\text{URC}^2$ exploits the distinction between aleatoric and epistemic uncertainty to enable *uncertainty-aware control*, identifying recurring failure modes and mitigating them via targeted reweighting and data augmentation. $\text{URC}^2$ provides a practical, reproducible recipe for building more trustworthy NLI systems under adversarial shift.

## 1 Introduction

Accuracy on static, i.i.d. benchmarks often masks brittleness under shift: models can be confidently wrong on adversarial inputs. The Adversarial NLI (ANLI) benchmark Nie et al. (2020) was designed precisely to surface these failures, yet most pipelines still treat uncertainty as a single scalar and optimize post-hoc calibration in isolation. This leaves two practical gaps: (i) distinguishing *aleatoric* ambiguity (data/label issues) from *epistemic* disagreement (model uncertainty), and (ii) acting on each with the right supervision rather than uniformly reweighting or purely post-hoc scaling.

We propose $\text{URC}^2$ (Uncertainty-Routed Curation & Calibration), a three-stage approach that (A) measures per-example uncertainty using a teacher ensemble and *decomposes* it into aleatoric and epistemic components, (B) *routes* examples by dominant source of uncertainty to a two-lane relabeling workflow (humans for ambiguity; an instruction-tuned LLM for confident model disagreement), and (C) *refreshes* the teachers with curated labels and instance weights, followed by lightweight temperature scaling. In short, $\text{URC}^2$ turns diagnosis into targeted supervision. Motivational example appears in Appx  AFig. 6)

Concretely, we instantiate the teacher ensemble with DeBERTa-v3-large, RoBERTa-large, and XLM-R-large and compute, per example, total uncertainty $H$, aleatoric $A$ (mean per-model entropy), and epistemic $E$ via a mutual-information style decomposition:

$$H = \mathbf{H}(\bar{p}), \quad A = \mathbf{E}_m[\mathsf{H}(p_m)], \quad E = H - A.$$

High-$A$ items are human-audited (relabel / keep-hard with down-weight / drop), while high-$E$ items—where models are individually confident yet disagree—are adjudicated by an instruction-tuned LLM with self-consistency checks. We then retrain and calibrate. Operational thresholds, prompts, selection heuristics, and optimizer details are deferred to the appendix.

**Definitions.** We report expected calibration error (ECE) and use standard temperature scaling (TS) Guo et al. (2017) on the concatenated ANLI dev split; full formulas are given in Sec. 4.

**Results on ANLI.** $URC^2$ improves probability quality without sacrificing accuracy: the ensemble's dev ECE drops from $0.249$ to $0.209$ pre-TS, and to **0.146** after temperature scaling, with accuracy held (§5). On the curated subset, epistemic mutual information collapses (large negative $\Delta$MI), indicating real disagreement removal rather than cosmetic smoothing. Corpus-level uncertainty shifts toward the low-aleatoric/low-epistemic quadrant after the teacher refresh. Full diagnostics, risk–coverage curves, and per-round breakdowns appear in the appendix.

**Contributions.**

1. *Uncertainty-driven supervision.* We operationalize an ensemble-based decomposition (aleatoric vs. epistemic) and use it to *route* examples to different supervision mechanisms.

2. *Human–LLM two-lane relabeling.* Ambiguity-heavy items go to human audit; confident model disagreements go to an instruction-tuned LLM with acceptance checks, yielding curated labels and per-example weights.

3. *Calibration with disagreement reduction.* On ANLI, $URC^2$ achieves strong dev calibration (ECE **0.146** with TS) at stable accuracy and consistent with targeted reduction of model-side disagreement, with favorable corpus-level uncertainty shifts.

**Terminology.** We refer to the end-to-end method as the $URC^2$ pipeline (three stages). Within Stage 2, the human–LLM relabeling workflow has two lanes: a *Human* lane for aleatoric-heavy cases and an *LLM* lane for epistemic-heavy cases. See figure 1 for more details. We use "pipeline" only for the full method and "workflow/lanes" only for Stage 2.

## 2 BACKGROUND AND PROBLEM STATEMENT

### 2.1 OVERVIEW OF ANLI

The ANLI dataset Nie et al. (2020) stress-tests NLI systems via a *human + model-in-the-loop* protocol where crowd workers craft hypotheses that fool a strong model yet remain clear to other annotators, across three adversarial rounds (R1–R3). It uses three labels: `entailment`, `neutral`, `contradiction`, with a roughly balanced distribution and a slight skew toward `neutral`, which can mask label-quality and disagreement issues. Premises are long passages paired with short adversarial hypotheses; dataset statistics and per-round uncertainty trends appear in Appx. A (Table 6, Fig. 5).

### 2.2 PROBLEM STATEMENT

Adversarial NLI benchmarks such as ANLI expose failures that accuracy alone cannot resolve: models are often confidently wrong, undermining trust in their outputs. These failures reflect two kinds of uncertainty: *aleatoric* ambiguity (multiple reasonable interpretations or noisy labels, where even an ideal model should remain uncertain) and *epistemic* disagreement (clear but adversarial inputs that make strong models diverge with high confidence, revealing blind spots). We seek to disentangle these sources and act on each appropriately by (i) quantifying aleatoric and epistemic uncertainty at the example level, (ii) using these signals for targeted supervision—routing ambiguous cases to human audit and disagreement-heavy cases to LLM adjudication, and (iii) showing that such routing improves calibration, label quality, and the corpus-level uncertainty distribution.

## 3 RELATED WORK

Adversarial NLI (ANLI) introduced a human + model-in-the-loop protocol that systematically surfaces shortcut reliance and blind spots that i.i.d. test sets can miss Nie et al. (2020), aligning with broader efforts in dynamic evaluation such as Dynabench Kiela et al. (2021). Earlier analyses revealed annotation artifacts enabling hypothesis-only baselines and brittle heuristics (e.g., HANS) Gururangan et al. (2018); Poliak et al. (2018); McCoy et al. (2019), while ChaosNLI Naeini et al. (2015) emphasized that annotator disagreement often reflects genuine ambiguity rather than mere

noise. These strands largely *diagnose* dataset pathologies; our focus is to *act* on them by routing examples to different forms of supervision based on their uncertainty type.

Estimating predictive uncertainty with deep ensembles and approximate Bayesian methods (e.g., MC Dropout) is practical and effective Lakshminarayanan et al. (2017); Gal & Ghahramani (2016). Following the standard view Kendall & Gal (2017), we decompose total uncertainty into *aleatoric* (data/label ambiguity) and *epistemic* (model uncertainty) components; in ensembles, the latter can be captured via a BALD-style mutual information term computed from model disagreement. Prior work predominantly uses such decomposition as a diagnostic; we instead *use* it to drive targeted curation: ambiguity-heavy items go to human audit, whereas confidently disagreeing items go to an instruction-tuned LLM for adjudication.

Calibration methods such as temperature scaling, Dirichlet, and beta calibration improve probability quality without altering accuracy Guo et al. (2017); Kull et al. (2019; 2017), yet calibration typically degrades under distribution shift Ovadia et al. (2019), and reported ECE on ANLI remains high even for strong models Hu et al. (2023). Rather than relying solely on post-hoc scaling, we first improve supervision quality through uncertainty-routed edits and instance weighting, and only then apply lightweight TS—yielding substantially lower ECE on ANLI. Model-agnostic debiasing methods for NLI (e.g., Liu et al. (2020)) similarly adjust training signals without changing architectures; $URC^2$ is complementary, using uncertainty decomposition and routing rather than artifact-specific reweighting, and focusing on calibration under adversarial shift.

A complementary literature addresses label quality and noisy-label learning Northcutt et al. (2021; 2019); Song et al. (2022). Consistent with these insights, our human lane corrects clear errors, drops irreparably noisy cases, and down-weights "keep-hard" items that remain ambiguous. In parallel, instruction-tuned LLMs have emerged as capable annotators and judges Ouyang et al. (2022); Gilardi et al. (2023); Zheng et al. (2023); our use is deliberately *selective*—restricted to high-epistemic items with confident model disagreement, with acceptance checks to avoid blanket relabeling. Finally, selective/conformal prediction offers deployment-time guarantees via abstention or set prediction Angelopoulos & Bates (2023); our training-time curation improves calibration and reduces disagreement, thereby strengthening the conditions under which such guarantees are most reliable.

**Positioning.** Active learning and noisy-label pipelines typically prioritize examples by model uncertainty or loss but do not distinguish aleatoric from epistemic causes Song et al. (2022); Northcutt et al. (2019). Methods such as ADDMU detect boundary cases via uncertainty signals but stop at diagnosis Yin et al. (2022). $URC^2$ uses an ensemble-based decomposition to route items to distinct interventions (human vs LLM), then folds decisions back through instance weighting and a teacher refresh before light post-hoc calibration, complementing deployment-time tools like selective/conformal prediction Angelopoulos & Bates (2023).

## 4 METHODOLOGY

### 4.1 PRELIMINARIES AND NOTATION

**Task and ensemble.** We study NLI with inputs $x = (p, h)$ (premise, hypothesis) and label $y \in \{1, \ldots, C\}$; for ANLI, $C=3$ with labels `entailment`, `neutral`, `contradiction`. To estimate uncertainty, we use an ensemble of $M$ fine-tuned *teachers* $\{\theta_m\}_{m=1}^{M}$, each producing posteriors $P_{\theta_m}(y \mid x)$. **Uncertainty decomposition.** Let the ensemble mean be

$$\bar{p}(y \mid x) = \frac{1}{M} \sum_{m=1}^{M} P_{\theta_m}(y \mid x). \tag{1}$$

We define total ($H$), aleatoric ($A$), and epistemic ($E$) uncertainty via entropies:

$$H(x) = - \sum_{y=1}^{C} \bar{p}(y \mid x) \log \bar{p}(y \mid x), \tag{2}$$

$$A(x) = \frac{1}{M} \sum_{m=1}^{M} \left[ - \sum_{y=1}^{C} P_{\theta_m}(y \mid x) \log P_{\theta_m}(y \mid x) \right], \tag{3}$$

$$E(x) = H(x) - A(x). \tag{4}$$

Here $H$ is the ensemble predictive entropy, $A$ is the mean per-model entropy capturing data/label ambiguity, and $E$ (their gap) quantifies reducible model disagreement (a mutual-information–style term). We also use the *epistemic share* $r(x) = E(x)/H(x)$ for $H(x)>0$.

**Confidence and calibration.** We use normalized entropy $\widehat{H}(x) = H(x)/\log C \in [0, 1]$ and define entropy-based confidence $c(x) = 1 - \widehat{H}(x)$. Calibration is measured via ECE with $B=10$ equal-width bins:

$$\text{ECE} = \sum_{b=1}^{B} \frac{|S_b|}{N} \big| \text{acc}(S_b) - \text{conf}(S_b) \big|. \tag{5}$$

where $S_b$ are the examples grouped into $B = 10$ equal-width bins based on their confidence scores. ECE quantifies the weighted average mismatch between confidence and accuracy across bins— lower is better, with ECE=0 indicating perfect calibration. Post-hoc temperature scaling (TS) Guo et al. (2017) rescales logits $z$ with a scalar $T$.

$$P_T(y \mid x) = \frac{\exp(z_y/T)}{\sum_{y'} \exp(z_{y'}/T)}. \tag{6}$$

where $T > 0$ is a learned scalar (fit by minimizing negative log-likelihood on a held-out set). For $T > 1$, predictions become softer (less confident); for $T < 1$, sharper. This preserves the argmax label but better aligns probabilities with true likelihoods. We instantiate $M=3$ diverse teachers—DeBERTa-v3-large, RoBERTa-large, XLM-R-large—to promote genuine epistemic variation (different inductive biases and pretraining). For each $x$ we compute $\bar{p}$, $H$, $A$, $E=H-A$, and $r=E/H$ from the teacher posteriors.

### 4.2 THE HUMAN–LLM RELABELING PIPELINE

URC$^2$ converts uncertainty diagnosis into targeted supervision through three coupled stages: (A) *measure & route* by dominant uncertainty type; (B) *human–LLM relabeling* in two disjoint lanes; (C) *retrain & calibrate* and refresh the ensemble. We iterate A→B→C until dev calibration plateaus. See figure 1 for detailed work flow.

#### 4.2.1 STAGE A — MEASURE & ROUTE (NO EDITS)

From the fixed teacher ensemble we compute $(H, A, E, r)$ and assign each example to *exactly one* lane by dominant uncertainty. An example is routed to the **epistemic-heavy** pool if *all* hold:

1. $r(x) = E(x)/H(x) \geq 0.35$    (disagreement dominates total uncertainty),
2. at least two teachers predict *different* argmax labels,
3. for the disagreeing teachers, the top–second probability margin $\geq 0.20$ (each is confident).

On ANLI train, $r(x)=0.35$ is near the 70th percentile of the epistemic-share distribution, so Lane L receives the top $\approx 30\%$ most epistemic-heavy examples. In our ensembles, strong model clashes typically have top–second probability gaps around 0.3–0.4, so we set the margin threshold to 0.20 to retain confident disagreements while discarding low-confidence cases. Otherwise, items with high $A(x)$ and low $r(x)$ are treated as **aleatoric-heavy** (ambiguous/noisy). Previously curated items (by either lane) are skipped in later rounds. Stage A *only* measures and routes; no labels or weights are changed here.

#### 4.2.2 STAGE B — TWO-LANE RELABELING (TARGETED SUPERVISION)

Each routed item is curated in *one disjoint lane*; items never cross lanes. Outputs are a possibly updated label and an instance weight $w_i$.

**Lane H (Human audit; aleatoric-heavy).** High-$A$, low-$r$ items typically reflect vague premises, underspecification, or label noise. Annotators choose among:

- *Relabel* (apply corrected label; weight $w=1.0$),
- *Keep-Hard* (retain original label but mark as ambiguous; $w=\alpha$, with $\alpha=0.3$),

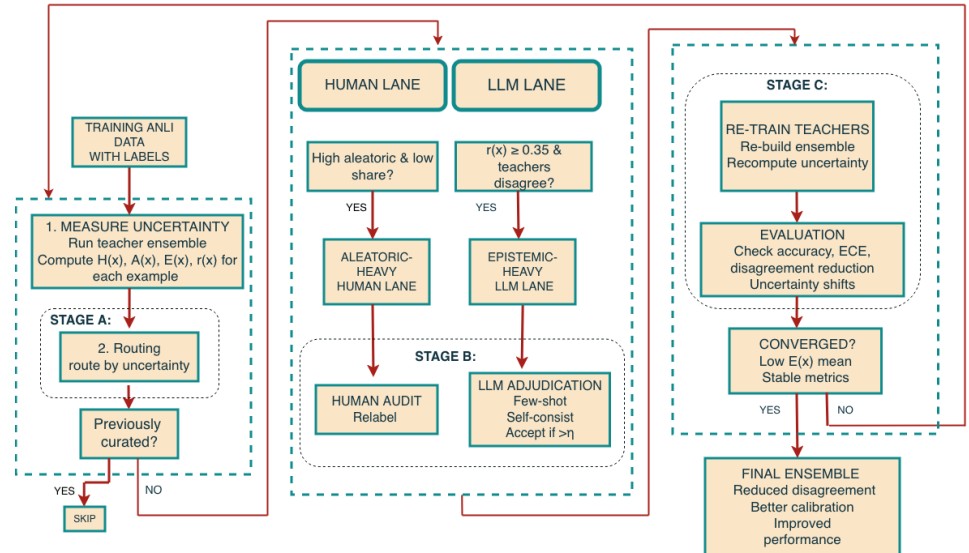

Figure 1: **URC$^2$ overview.** Stage A routes by uncertainty; Stage B applies a two-lane Human–LLM relabeling; Stage C retrains and calibrates, closing the loop.

- *Drop* (remove from training; $w$=0.0).

This preserves informative border cases while preventing ambiguity from dominating, and removes irredeemably ill-posed items.

**Lane L (LLM adjudication; epistemic-heavy).** When teachers are individually confident yet disagree (high $r(x)$, low $A(x)$, and a top–second margin $\geq 0.20$), the example is sent to an instruction-tuned NLI LLM to make a single, consistent call. For each routed item we generate $n_{\text{try}}$=3 stochastic completions and accept an LLM label only if one class attains at least 70% of the votes *and* that label differs from the original ANLI label; otherwise we keep the original label. Accepted LLM labels are given full weight ($w$=1.0) and their indices stored in $\mathcal{S}_{\text{chg}}$ for later $\Delta$MI analysis, so the intervention is limited to a small epistemic-heavy slice of the data rather than relabeling ANLI wholesale. Configuration and human-audit details for Lane L are summarized in Appx. C.2. This separation lets Lane H treat ambiguous, aleatoric-heavy items conservatively (keep, down-weight, or drop), while Lane L is reserved for clear-but-disagreed, epistemic-heavy examples where a single defensible label is expected. In small sweeps, choosing $\alpha$ anywhere in $[0.2, 0.5]$ produced similar calibration trends, so we fix $\alpha$=0.3 for all reported experiments.

### 4.2.3 STAGE C — RETRAIN, RECALIBRATE, REFRESH

Curated labels and weights from both lanes are folded back into *all* teachers to reduce disagreement and improve probability quality. Inputs. (i) Curated labels $\{y_i^{\text{clean}}\}$ and instance weights $\{w_i\}$ from Stage B; (ii) same splits/tokenization; (iii) the original fine-tuning recipe. See Appx C.1, Table 10 for training details.

**Procedure.** We (i) retrain teachers with weighted cross-entropy $\mathcal{L} = \sum_i w_i \, \ell(f_\theta(x_i), y_i^{\text{clean}})$ and early stopping on dev NLL, (ii) rebuild the ensemble and recompute $\bar{p}$, $H$, $A$, $E$, $r$, and (iii) fit a temperature $T$ on dev (by NLL) and report accuracy and ECE with/without TS (argmax unchanged).

**Diagnostics, stopping, and optimization.** After each refresh we evaluate progress on three axes. On the training set we track calibration via ECE; on the development set we track accuracy together with post–temperature-scaling ECE. We also monitor the uncertainty landscape via corpus means of total ($H$), aleatoric ($A$), and epistemic ($E$) uncertainty, the expected epistemic share $\mathbf{E}[E/H \mid H > 0]$, and the mass of examples in each $A \times E$ quadrant. We continue iterating the A→B→C loop while dev ECE keeps decreasing and the share of high-$E$ examples contracts; once both trends plateau, we stop. Training details appear in Appx. C.4.

Table 1: Post-hoc calibration baselines on ANLI dev (no curation). Ensemble uses mean of teacher posteriors.

| Method | Acc | Ent-ECE | MaxP-ECE | NLL | Brier |
|---|---|---|---|---|---|
| Uncalibrated | 0.514 | 0.202 | 0.169 | 0.996 | 0.620 |
| Temperature Scaling | 0.514 | 0.251 | **0.122** | 0.951 | 0.595 |
| Histogram Binning (OVR) | 0.515 | 0.252 | 0.118 | 0.952 | 0.595 |

Table 2: Comparison of best post-hoc method (OVR, no curation) vs. URC$^2$ with temperature scaling on ANLI dev.

| Method | Acc | Ent-ECE | MaxP-ECE | NLL |
|---|---|---|---|---|
| Best post-hoc (OVR) | 0.515 | 0.252 | **0.118** | 0.952 |
| URC$^2$ (+TS) | **0.581** | **0.146** | 0.120 | **0.884** |

## 5 EVALUATION PROTOCOL AND METRICS

We evaluate on ANLI dev using Ent-ECE, MaxP-ECE, NLL, and Brier score, with confidence defined as in Sec. 4. We report train and dev metrics before/after curation and temperature scaling, using the same splits, tokenization, and ensemble setup as in the Method section. ANLI dev is used only for early stopping and fitting the temperature parameter; routing thresholds and human weights are fixed *a priori* and never tuned on dev calibration metrics.

**Visual analyses.** We include: (i) per-round summaries of epistemic (MI) and aleatoric uncertainty with 95% confidence intervals (Fig. 5 in Appx. A), showing higher epistemic in later rounds (R3) and higher aleatoric in earlier ones (R1); and (ii) scatter plots of epistemic vs. aleatoric uncertainty on random subsamples (Fig. 5 in Appx. A), revealing distinct regimes (aleatoric-dominated, epistemic-dominated, mixed).

### 5.1 CALIBRATION AND ACCURACY METRICS

We use normalized entropy $\widehat{H}(x) = H(x)/\log C \in [0, 1]$ and define confidence $c(x) = 1 - \widehat{H}(x)$. ECE computed over 10 equal-width confidence bins. We report train ECE (pre/post curation) and dev ECE after post-hoc temperature scaling (TS), which minimizes dev NLL but leaves accuracy (argmax labels) unchanged. Reliability diagrams (Fig. 2, 3) visualize the bin-wise gaps.

### 5.2 CALIBRATION BASELINES (NO CURATION)

We benchmark standard post-hoc calibrators on the uncurated ensemble (mean of teacher posteriors) using Ent-ECE, MaxP-ECE, NLL, and Brier scores. As shown in Table 1, temperature scaling (TS) and histogram binning improve MaxP-ECE and NLL over the raw ensemble, but Ent-ECE remains high. In contrast, URC$^2$—which modifies supervision (Stage B) then retrains and calibrates (Stage C)—achieves substantially lower Ent-ECE (0.146 vs. 0.251/0.252 for TS/histogram on uncurated data), while also improving MaxP-ECE and proper scores (Tables 2 and 11).
This gap shows that correcting labels in disagreed-upon examples and retraining teachers improves probability quality beyond post-hoc methods alone. Post-hoc calibration helps, but URC$^2$ (refresh + TS) is consistently better, as it fixes supervision before calibrating rather than just rescaling uncurated outputs.

To assess calibration improvements, we analyze corpus-level changes pre/post curation: (i) **Means**: Track $\mathbf{E}[H]$, $\mathbf{E}[A]$, $\mathbf{E}[E]$, and epistemic share $\mathbf{E}[r(x) \mid H(x){>}0]$. Expect reduced $\mathbf{E}[E]$ from less disagreement and slightly lower $\mathbf{E}[A]$ from fewer noisy items. (ii) **Quadrant shares**: Using fixed high/low $A$ and $E$ cutoffs, measure mass in high-$E$/low-$A$ (disagreement), high-$A$/low-$E$ (ambiguity), both-high, and both-low. Success shifts mass to both-low, reducing high-$E$. We iterate until: (i) train ECE drops with stable accuracy; (ii) dev ECE improves post-TS with competitive accuracy; and (iii) $\mathbf{E}[E]$ and high-$E$ mass decrease. Stop when calibration and disagreement metrics stabilize.

Table 3: Dev calibration of the teacher ensemble .

| Model | Acc (pre-TS) | ECE (pre-TS) | $T_{\text{NLL}}^{\star}$ | ECE@$T_{\text{NLL}}^{\star}$ |
|---|---|---|---|---|
| Ensemble (DeBERTa-v3, RoBERTa, XLM-R) | 0.581 | 0.209 | 0.53 | **0.146** |

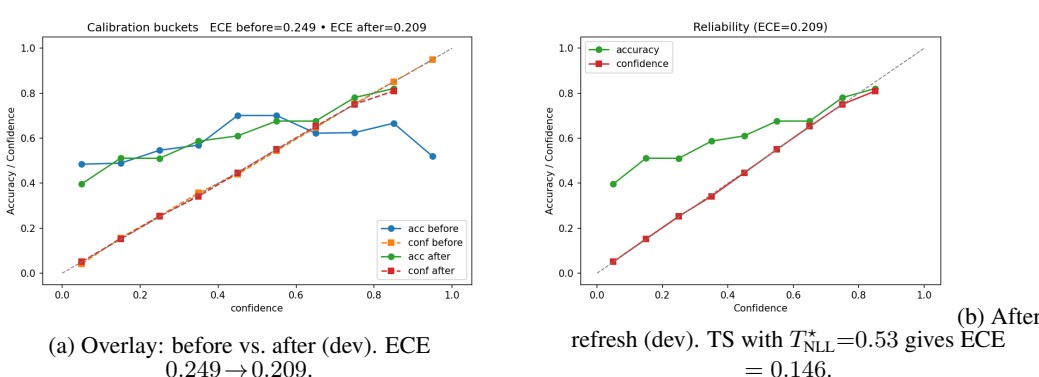

(a) Overlay: before vs. after (dev). ECE $0.249 \rightarrow 0.209$.

(b) After refresh (dev). TS with $T_{\text{NLL}}^{\star}=0.53$ gives ECE $= 0.146$.

Figure 2: Reliability of the teacher ensemble on ANLI dev (pre-TS and post-refresh).

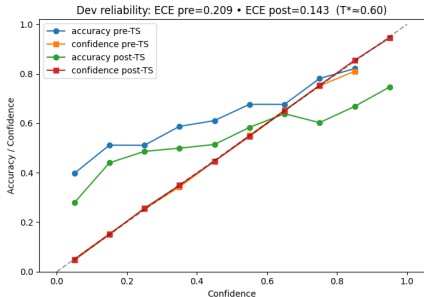

Figure 3: Reliability on ANLI dev before/after TS ($T_{\text{NLL}}^{\star} = 0.53$). ECE $0.209 \rightarrow$ **0.146** at unchanged accuracy; TS mainly lowers confidence in the highest bins.

# 6 RESULTS

We evaluate $URC^2$ on ANLI with two goals: (i) **probability quality** on dev (accuracy, ECE) and (ii) **disagreement resolution** on train (epistemic MI and the overall uncertainty landscape). We first report ensemble calibration on dev, then quantify what changed due to curation (footprint and $\Delta$MI), and finally show corpus-level shifts after the all-teacher refresh.

After the refresh, the dev reliability curve moves toward the diagonal in the mid-confidence region, reducing pre-TS ECE from $0.249$ to $0.209$. A light ECE-tuned temperature $T_{\text{ECE}}^{\star} = 0.53$ brings ECE to $0.146$ while accuracy remains $0.581$ (Table 3; Figs. 2, 3).

## 6.1 CURATION FOOTPRINT AND RESOLVED DISAGREEMENT

The two-lane correction makes a focused intervention: a small human audit (500 items; 99 relabeled, 11 dropped, the remainder down-weighted) complements LLM adjudication with 6,152 accepted edits. On the curated set $\mathcal{S}_{\text{chg}}$ (indices with accepted label edits), ensemble epistemic MI falls from $0.4663$ to $0.0739$ ($\Delta$MI $= -0.3924$), indicating that disagreement is largely eliminated where supervision actually changed. In Fig. 4, the decile panel places post-curation MI in a narrow band even for the hardest deciles, while the $\Delta$MI histogram concentrates its mass at $\leq 0$; together, these patterns are consistent with adjudication removing model-side disagreement on edited rows—an effect that post-hoc temperature scaling cannot produce. As a sanity check, a single-teacher pilot reduced train ECE from $0.235$ to $0.105$ at fixed accuracy (dev ECE $0.208 \rightarrow 0.177$ with TS), underscoring that gains stem from cleaner supervision rather than accuracy gaming; per-teacher breakdowns appear in App. D, Table 11.

Table 4: Curation footprint and MI reduction on curated items.

| Item | Value | Note |
|---|---|---|
| Human audit (aleatoric-heavy) | 500 | Relabel 99 (19.8%), drop 11 (2.2%); others retained/down-weighted. |
| LLM adjudication (accepted changes) | **6,152** | Applied at $w{=}1.0$. |
| MI on curated set $\mathcal{S}_{\text{chg}}$ | $0.4663 \rightarrow \mathbf{0.0739}$ | $\Delta\text{MI} = -\mathbf{0.3924}$ (after$-$before). |

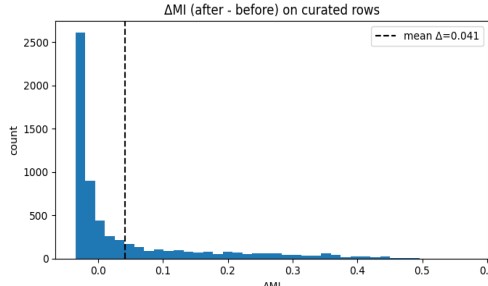

(a) MI by decile (curated rows).  (b) Distribution of $\Delta$MI (curated rows).

Figure 4: Resolved disagreement on curated rows. (a) Deciles formed by *pre-adjudication* MI; after adjudication, MI lies in a narrow band across all deciles ($< 0.085$). (b) Per-item $\Delta$MI (after$-$before) concentrates at $\leq 0$ with a long positive tail; the small right-shifted mean is driven by few outliers.

## 6.2 $\Delta$MI ON CURATED ROWS AND SELECTIVE PREDICTION (BRIEF)

On the rows actually edited during curation ($\mathcal{S}$; tensors standardized to $(N, 3)$ with RoBERTa class-order alignment), the change in ensemble epistemic MI is large and localized: $|\mathcal{S}| = 6{,}152$, mean $\Delta$MI (after$-$before) $=-\mathbf{0.247}$, median $=-\mathbf{0.256}$. This establishes that adjudication resolves model-side disagreement where labels changed (full bars/hist in Appx. E, Fig. 8). Complementing this, selective prediction improves across the coverage range after the refresh: E-AURC decreases from $0.399$ to $0.321$ with entropy confidence and from $0.390$ to $0.318$ with max-probability confidence, indicating better ranking of easy vs. hard examples. Because temperature scaling is monotone, these gains are attributable to the refresh rather than post-hoc calibration (see Appx. D, Fig. 7 for the full risk–coverage panels).

Table 5: Uncertainty landscape on full train after refresh.

| Quadrant shares | | Corpus means | |
|---|---|---|---|
| Quadrant | Share | Metric | Value |
| Low-A / Low-E | **0.700** | Mean $H$ | 0.393 |
| High-A only | 0.050 | Mean $A$ | 0.358 |
| High-E only | 0.050 | Mean $E$ | **0.035** |
| Both-high | 0.200 | $E/H$ | 0.050 |

To test if URC$^2$ reduces to an "LLM relabels everything" strategy, we ran the same Lane L adjudicator *without* uncertainty routing on a random 20,000-example subset of ANLI train, using $n_{\text{try}}{=}3$ and the 70% self-consistency rule. The LLM accepted 12,617 items and changed 5,340 labels (26.7% of the subset). Within this accepted set, the label distribution shifts from 41.6/34.8/23.6% to 73.6/0.4/26.0% (ENT/NEU/CON), effectively eliminating `neutral` cases. By contrast, URC$^2$ keeps aleatoric-heavy items in Lane H (often kept or down-weighted) and restricts Lane L to epistemic-heavy disagreements (see Appx. C.3).

## 7 ABLATION STUDIES

We consider four ablations/baselines: (i) a temperature-scaling (TS) baseline on dev with *no* curation, (ii) the causal change in epistemic disagreement on the actually edited rows $\mathcal{S}$, (iii) a naive

"LLM relabels everything" baseline, and (iv) the post-refresh uncertainty landscape. Full numeric tables and plots are deferred to the appendix.

**TS-only (no curation) on dev.** With labels untouched (pre-refresh dev), the teachers and their mean ensemble provide the calibration baseline. Applying NLL-tuned TS makes a *pure* likelihood adjustment: dev NLL improves from $0.874$ to $0.846$ at $T^\star = 1.50$, but calibration worsens (Ent-ECE $0.340 \to 0.453$; MaxP-ECE $0.040 \to 0.096$). This expected trade-off (we optimize NLL, not ECE) shows that the calibration gains of URC$^2$ cannot be attributed to TS alone; they come from *curation+refresh* rather than post-hoc scaling.

**$\Delta$MI on curated rows $\mathcal{S}$.** We measure the change in ensemble epistemic mutual information (MI) on the rows that actually changed during curation ($\mathcal{S}$; RoBERTa class order aligned; tensors standardized to $(N, 3)$). The effect is large and localized: $|\mathcal{S}| = 6{,}152$, with mean $\widehat{\Delta}$MI (after $-$ before) $= -\mathbf{0.247}$ and median $-\mathbf{0.256}$. The corresponding bar plot and histogram are shown in Appx. E, Fig. 8, and confirm that adjudication resolves model-side disagreement where labels are edited.

**Naive LLM-for-all baseline.** As a naive baseline we run the Lane L adjudicator *without* routing on 20,000 ANLI train examples ($n_{\text{try}}=3$, 70% rule). It accepts 12,617 items, changes 5,340 labels (26.7%), and shifts the accepted label mix from 41.6/34.8/23.6% to 73.6/0.4/26.0% (E/NEU/C), effectively collapsing almost all `neutral` cases; URC$^2$ avoids this by routing aleatoric-heavy items to Lane H and restricting Lane L to epistemic-heavy disagreements (Appx. C.3).

**Uncertainty landscape after refresh.** Post-refresh, most training examples concentrate in low- or single-source uncertainty regimes. Using balanced percentile cutoffs (70th/70th) for high aleatoric ($A$) and high epistemic ($E$), the corpus splits as: $0.400$ low-$A$/low-$E$, $0.300$ high-$A$ only, $0.300$ high-$E$ only, and essentially $0.000$ jointly high. This pattern is consistent with disagreement being resolved rather than masked. Sensitivity to absolute thresholds and additional diagnostics are deferred to the appendix (Appx. G, Table 17).

## 8 DISCUSSION

Refreshing teachers on curated supervision improves reliability without changing decisions. On ANLI dev, pre-TS ECE drops from $0.249$ to $0.209$; an ECE-tuned temperature brings it to $0.146$ with accuracy fixed at $0.581$ (risk–coverage panels in Appx. D.2). Where supervision actually changed, disagreement is resolved: on $\mathcal{S}_{\text{chg}}$ (6,152 edits), ensemble epistemic MI falls from $0.4663$ to $0.0739$ ($\Delta$MI $= -0.3924$; full bars/hist in Appx. E). Corpus-wide after the refresh, uncertainty mass concentrates in low-uncertainty regions (both-low $0.578$) with a small epistemic share ($\sim 5\%$), consistent with the curated-slice gains; absolute $2{\times}2$ thresholds and sensitivity appear in Appx. G. The effect comes from splitting uncertainty (aleatoric vs. epistemic), routing to the right supervision (human vs. LLM), and performing a single refresh; per-teacher dev diagnostics are in Appx. D.

## 9 CONCLUSION

We presented an uncertainty-routed curation and calibration pipeline that *explicitly distinguishes* aleatoric ambiguity from epistemic disagreement and addresses each with the right tool—human audit for ambiguity and instruction-tuned LLM adjudication for disagreement—followed by a full-teacher refresh and temperature scaling. On ANLI, this yields (i) dev ECE improvements from $0.249{\to}0.209$ (pre-TS) and to $0.146$ with TS at unchanged accuracy $0.581$ (Fig. 2, Table 3); (ii) large, targeted disagreement removal on curated rows ($\Delta$MI$=-0.3924$ over 6,152 changed items; Table 4); and (iii) a corpus-level shift toward low-uncertainty regions with a small epistemic share (Table 5). Beyond improving a single metric, URC$^2$ turns uncertainty *diagnosis* into *actionable supervision*, delivering calibrated probabilities and causally reducing model-side disagreement at scale—a general, reproducible recipe for reliable, LLM-in-the-loop dataset improvement.

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

# A  ADDITIONAL BACKGROUND ON ANLI

*Purpose.* This section supports Sec. 2 with per-round uncertainty trends and dataset statistics.

ANLI was released in three adversarial rounds (R1–R3). Each round used progressively stronger models as "adversaries," requiring annotators to construct increasingly subtle hypotheses to break them. This results in systematic differences:

- **R1**: Smaller, noisier, with many underspecified or ambiguous hypotheses.

- **R2**: More balanced, but still contains artifacts and label disagreement.

- **R3**: By far the largest, adversarially sharper, with higher epistemic uncertainty (models disagree while annotators converge).

*What this shows.* Tab. 6 summarizes splits and collection protocol for quick reference.

Table 6: **ANLI dataset statistics** by adversarial round (R1–R3).

| Aspect | Details |
| --- | --- |
| Train size (R1 / R2 / R3) | 16,946 / 45,460 / 100,459 |
| Dev size (R1 / R2 / R3) | 1,000 / 1,000 / 1,200 |
| Test size (R1 / R2 / R3) | 1,000 / 1,000 / 1,200 |
| Label set | entailment, neutral, contradiction |
| Collection method | Adversarial, human-in-the-loop (R1–R3) |

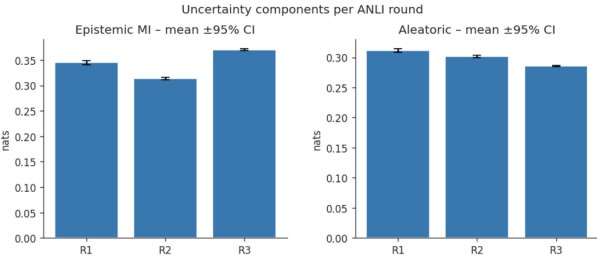

Figure 5: **Uncertainty by round.** Mean (±95% CI) epistemic MI (left) and aleatoric (right) across R1–R3.

**Input lengths and structure.**  Premises in ANLI are typically long, multi-sentence passages, while hypotheses are short and cue-sensitive (negations, quantifiers, numerals). This asymmetry raises two practical issues: (i) long premises may be truncated under short sequence limits, removing needed evidence; and (ii) short hypotheses concentrate weight on a few tokens (e.g., "not," "three," "most"), encouraging superficial cues and high-variance predictions. Representative examples illustrating these patterns appear in Appx. F, Table 16.

# B  EXTENDED RELATED WORK

## B.1  ANLI CASE STUDY

*What this shows.* Fig. 6 demonstrates the effect of URC$^2$: routing by uncertainty sends the *epistemic* item(example A) to LLM adjudication, collapsing disagreement and concentrating mass on CONTRADICTION $(0.07, 0.34, 0.59) \rightarrow (0.05, 0.08, 0.87)$; the *aleatoric* item(Example B) goes to human audit and remains deliberately cautious while leaning NEUTRAL, shifting $(0.36, 0.37, 0.27) \rightarrow (0.30, 0.41, 0.29)$ without forcing artificial agreement.

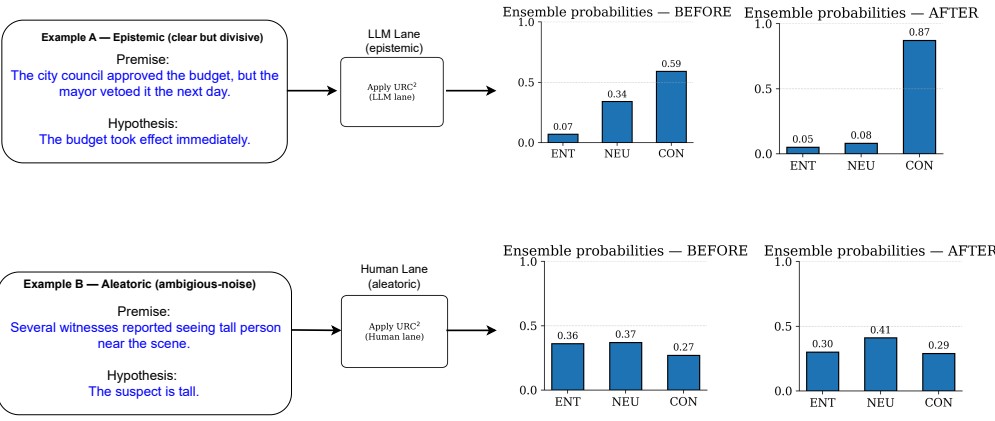

Figure 6: Two ANLI examples with ensemble probabilities before/after applying URC$^2$.

## B.2 ADVERSARIAL NLI AND DYNAMIC EVALUATION

ANLI's human + model-in-the-loop procedure Nie et al. (2020) iteratively elicits examples that fool strong models while staying clear to humans, complementing dynamic benchmarking such as Dynabench Kiela et al. (2021). Prior analyses exposed annotation artifacts enabling hypothesis-only baselines Gururangan et al. (2018); Poliak et al. (2018); McCoy et al. (2019). ChaosNLI Nie et al. (2020)showed annotator disagreement often reflects genuine ambiguity, motivating ambiguity-aware treatment. We build directly on this line by turning diagnosis into intervention via uncertainty-routed curation.

## B.3 PREDICTIVE UNCERTAINTY: ESTIMATORS AND DECOMPOSITION

Approximate Bayesian methods and deep ensembles provide practical uncertainty in modern NLP. MC Dropout Gal & Ghahramani (2016) and deep ensembles Lakshminarayanan et al. (2017) are widely used. Following Kendall & Gal (2017), we split uncertainty into *aleatoric* (data/label ambiguity) and *epistemic* (model) components. For ensembles, epistemic uncertainty can be quantified by mutual information (BALD) Houlsby et al. (2011); Smith & Gal (2018). Alternatives include Dirichlet Prior Networks Malinin & Gales (2018), evidential models, and temperature/Dirichlet/beta calibrations. Surveys Hu et al. (2023); Shorinwa et al. (2024) catalog this space and highlight decomposition's utility under shift. Unlike methods such as ADDMU Yin et al. (2022) that primarily *detect* far-boundary points, we *route* examples by uncertainty type into distinct supervision lanes and retrain.

## B.4 CALIBRATION UNDER DISTRIBUTION SHIFT

Post-hoc calibration (temperature scaling Guo et al. (2017), Dirichlet Kull et al. (2019), beta Kull et al. (2017)) aligns confidence with empirical accuracy without changing argmax labels. Calibration often degrades under shift Ovadia et al. (2019); large ECE values have been reported on ANLI even for strong models Hu et al. (2023). We therefore improve supervision first via routing+curation, then apply light TS; empirically this yields lower ECE than post-hoc scaling alone.

## B.5 LABEL QUALITY, NOISY LABELS, AND INSTANCE WEIGHTING

Label issues are pervasive in large datasets Northcutt et al. (2021). Methods include confident learning Northcutt et al. (2019), sample selection, and instance weighting Song et al. (2022). Noisy-label surveys Song et al. (2022) and NLP benchmarks Wang et al. (2023) argue for treating ambiguity differently from clear errors. Our human lane mirrors this: correct clear mistakes, drop irrecoverable items, and down-weight "keep-hard" ambiguous cases.

Table 7: **Lane L: LLM adjudication configuration** (model, prompting, parsing, acceptance, selection).

| Component | Setting |
|---|---|
| Model | Meta-Llama-3-8B-Instruct (local); 8-bit quantization (BitsAndBytes); FP16 |
| Prompt | 3 anchor exemplars (ENT/NEU/CON) + 6 retrieved demos (see Tab. 9); strict JSON output |
| Preprocessing | ASCII cleaning; truncate: premise 180 chars, hypothesis 120 chars |
| Sampling | `do_sample`=True, $T$=0.7, top_p= 0.9, max_new_tokens= 256, $n\_try$=3 |
| Parsing | Regex for `{"label":"ENT"}` / `"NEU"` / `"CON"`; invalid $\Rightarrow$ no change |
| Acceptance | Apply only if predicted label $\neq$ original; if applied, set instance weight $w$=1.0 and log to $\mathcal{S}_{\text{chg}}$ |
| Selection | Hard pool: top 6,000 by $s(x)=H(\bar{p}(\cdot|x)) - \big(p_{(1)}(x) - p_{(2)}(x)\big)$ (confident disagreements) |
| Artifacts | `train_labels_llama_str.pkl`, `train_labels_llama_bin.pkl`, `llama_patched_idx.npy` |

## B.6 LLMS AS ANNOTATORS AND JUDGES

Instruction-tuned LLMs (e.g., InstructGPT/FLAN/T0) show strong agreement with human labels Ouyang et al. (2022); Wei et al. (2021); Sanh et al. (2022); Gilardi et al. (2023) and serve as automatic judges Zheng et al. (2023). We invoke an instruction-tuned LLM *only* on high-epistemic items where teacher models are confident yet disagree, with acceptance checks before folding decisions back into training—contrasting with blanket relabeling Pavlovic & Poesio (2024).

## B.7 SELECTIVE/CONFORMAL PREDICTION AND RISK–COVERAGE

Selective prediction and conformal prediction Vovk et al. (2005); Angelopoulos & Bates (2023) provide coverage guarantees by abstaining or outputting sets. They are deployment-time safeguards rather than supervision improvers. We report risk–coverage and E-AURC to quantify ranking quality before/after refresh; improvements mirror our calibration gains and complement conformal methods.

## B.8 POSITIONING AND SUMMARY

Across strands—adversarial benchmarking, uncertainty estimation, calibration, noisy-label learning, and LLM annotation—most prior work diagnoses failure or calibrates post hoc in isolation. Our contribution is to *integrate* them: use ensemble-based decomposition to identify *why* an example is uncertain (aleatoric vs. epistemic), *route* it appropriately (human vs. LLM), and *refresh* teachers before minimal post-hoc scaling. This yields lower ECE and measurable disagreement reduction on the edited slice, while improving the corpus-level uncertainty mix.

# C  LANE-L CONFIGURATION (PROMPTING, RETRIEVAL, ACCEPTANCE)

*Scope.* This section specifies the Lane L (LLM adjudication) setup end-to-end: model/prompting, demo retrieval, selection/acceptance, and the training recipe used to refresh teachers after curation. All components here correspond to the same scenario and are used together in our experiments.

*What this shows.* Tab. 7 lays out the exact knobs used for LLM adjudication so results are reproducible.

## C.1 DEMO RETRIEVAL DETAILS

*Why this matters.* Tabs. 8–9 document the retrieval setup that conditions the adjudicator with in-domain demos.

Table 8: Retrieval encoder specs for demo selection (all-mpnet-base-v2).

| Property | Detail |
|---|---|
| Model | sentence-transformers/all-mpnet-base-v2 |
| Embedding dim | 768 |
| Max input length | $\sim$512 tokens |
| Pooling | Mean pooling; L2-normalized |
| Similarity | Cosine similarity |
| Use in pipeline | Encode `premise [SEP] hypothesis` for queries & pool |

Table 9: Demo retrieval configuration used by Lane L prompting.

| Component | Setting |
|---|---|
| Query construction | `premise [SEP] hypothesis`; clean+trim (180/120 chars) |
| Encoder | all-mpnet-base-v2; cosine over L2-normalized embeddings |
| Anchors / retrieved shots | 3 anchors + top-$K$=6 nearest neighbors |
| Pool | Labeled ENT/NEU/CON premise–hypothesis pairs |
| Prompt assembly | 3 anchors + 6 demos + query; enforce JSON schema |
| Purpose | Improve LLM consistency via topic/structure similarity |

## C.2 HUMAN AUDIT OF LANE-L RELABELS

To sanity-check Lane-L adjudications, we randomly sampled 121 training examples from the high-epistemic pool where Lane L changed the label. For each item we recorded (i) the original ANLI label, (ii) the Lane-L LLM label, and (iii) an independent human NLI judgment (`entailment` / `neutral` / `contradiction`).

After mapping labels to three classes, the new human annotator agreed with the LLM on 85/121 examples (70.2%) and with the original ANLI label on 85/121 (70.2%). In the remaining 36/121 cases (29.8%), the human disagreed with both; 14 of these were instances where the human chose `neutral` while both ANLI and the LLM chose `entailment` or `contradiction`. We observed no systematic class bias in the LLM's decisions.

## C.3 LLM-FOR-ALL BASELINE ON A 20K RANDOM SUBSET

For the baseline described in Sec. 7, we sampled 20,000 ANLI training examples uniformly at random and ran the Lane L adjudicator without any uncertainty routing. The LLM accepted 12,617 examples, of which 5,340 labels were changed (42.3% of accepted; 26.7% of the 20k subset). Among the accepted examples, the original label proportions were 41.6% `entailment`, 34.8% `neutral`, and 23.6% `contradiction`; after LLM relabeling these became 73.6%, 0.4%, and 26.0%, respectively, indicating that nearly all `neutral` items are forced into a hard `entailment` or `contradiction` label.

## C.4 TRAINING RECIPE FOR TEACHER REFRESH

*What this shows.* Tab. 10 gives the post-curation optimization settings used across teacher models.

Table 10: **Training recipe (all teachers)** used after Lane L/Human curation.

| Component | Setting |
|---|---|
| Optimizer | AdamW (wd=0.05, $\beta_2$=0.98, $\epsilon$=$10^{-8}$) |
| Stability | Grad clip = 1.0; label smoothing = 0.05 |
| Schedule | Warmup 6–10% $\rightarrow$ cosine decay |
| Precision | FP16 with dynamic loss scaling |
| Batching | Small per-GPU batch + gradient accumulation |
| Memory | Gradient checkpointing |
| Init freeze | Optional: freeze embeddings for first 300–1000 steps |
| Early stopping | Dev NLL |
| Repro | Seeds/hardware in released configs |

Table 11: ANLI dev calibration by model and ensemble (entropy-based confidence, $B=10$ bins).

| Model | Acc (pre-TS) | ECE (pre-TS) | $T^\star$ | ECE@$T^\star$ | $\Delta$ECE |
|---|---|---|---|---|---|
| DeBERTa-v3-large | 0.664 | 0.148 | 0.85 | 0.104 | $-0.045$ |
| RoBERTa-large | 0.533 | 0.135 | 1.07 | 0.123 | $-0.011$ |
| XLM-R-large | 0.492 | 0.150 | 1.11 | 0.144 | $-0.006$ |
| **Ensemble** | 0.581 | 0.209 | 0.53 | **0.146** | $-0.067$ |

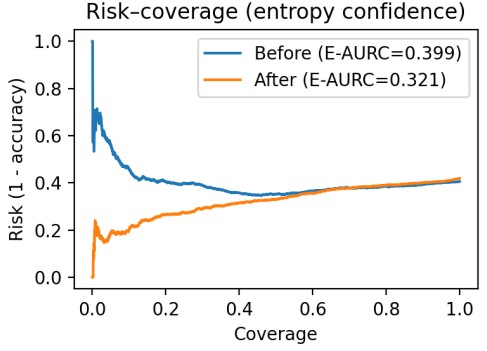

(a) Entropy confidence (E-AURC $0.399 \rightarrow 0.321$).

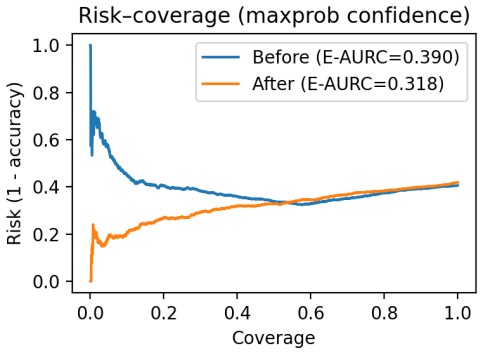

(b) Max-probability confidence (E-AURC $0.390 \rightarrow$ 0.318).

Figure 7: Selective prediction on ANLI dev. Post-refresh models achieve lower risk across coverage; the largest gap is at low coverage.

Table 12: URC$^2$ ensemble (post-refresh) on ANLI dev: metrics with 95% CIs.

| Metric | Point | 95% CI |
|---|---|---|
| Accuracy | 0.581 | [0.564, 0.598] |
| Ent-ECE | 0.209 | [0.194, 0.228] |
| MaxP-ECE | 0.120 | [0.104, 0.138] |
| NLL | 0.884 | [0.856, 0.913] |
| Brier | 0.543 | [0.525, 0.561] |

# D  ADDITIONAL DEV DIAGNOSTICS

## D.1  PER-MODEL DEV CALIBRATION

*Reading Table 11.* DeBERTa-v3-large attains the highest accuracy (0.664) and the lowest post-TS ECE (0.104). The ensemble shows the largest calibration gain from TS ($\Delta$ECE$= -0.067$), consistent with Fig. 2 where post-refresh accuracy exceeds confidence in mid bins; its $T^\star=0.53$ *sharpens* probabilities to close those gaps. RoBERTa and XLM-R require mild *softening* ($T^\star>1$) and see smaller ECE changes. Overall, TS improves probability quality without affecting accuracy, and the ensemble—used for uncertainty decomposition and routing—achieves competitive post-TS calibration (ECE 0.146) while benefiting most from re-calibration.

## D.2  SELECTIVE PREDICTION (RISK–COVERAGE)

*What this shows.* Fig. 7 compares risk across coverage before/after refresh for two confidence proxies.

## D.3  DEV METRICS WITH CIS (ENSEMBLE)

*What this shows.* Tab. 12 reports point estimates with 95% CIs for the post-refresh ensemble.

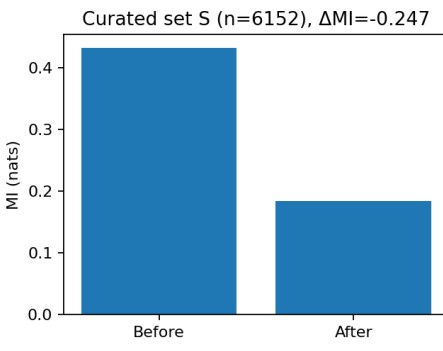

(a) MI before vs. after (bars)

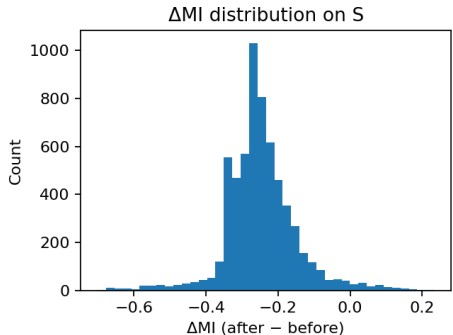

(b) Histogram of ΔMI (after−before)

Figure 8: **Curated rows $\mathcal{S}$ (full train).** Left: MI before vs. after (bars). Right: distribution of ΔMI (after−before). The shift left indicates disagreement reduction on edited items.

### D.4 TS-ONLY ABLATION ON ANLI DEV

This subsection provides the full numerical results for the TS-only baseline on ANLI dev discussed in Sec. A. We report (a) per-model pre-TS baselines without any curation, (b) ensemble behavior before vs. after NLL-tuned temperature scaling, and (c) the resulting uncertainty quadrants.

Table 13: TS-only ablation on ANLI dev — (a) Per-model baselines (pre-TS; no curation).

| Model | Acc | Ent-ECE | MaxP-ECE | NLL | Brier |
|---|---|---|---|---|---|
| DeBERTa-v3 | 0.719 | 0.177 | 0.220 | 1.256 | 0.497 |
| RoBERTa-large | 0.414 | 0.231 | 0.393 | 2.102 | 0.926 |
| XLM-R-large | 0.562 | 0.266 | 0.344 | 1.867 | 0.760 |

Table 14: TS-only ablation on ANLI dev — (b) Ensemble before vs. after TS (NLL-tuned, $T^{\star}=1.50$).

| Metric | Before | After (TS, $T^{\star}=1.50$) | Δ |
|---|---|---|---|
| Ent-ECE | 0.340 | 0.453 | 0.113 |
| MaxP-ECE | 0.040 | 0.096 | 0.056 |
| NLL | 0.874 | **0.846** | −0.028 |
| Brier | 0.452 | 0.471 | 0.019 |

Table 15: TS-only ablation on ANLI dev — (c) Uncertainty quadrants after refresh (true 2×2).

| Quadrant shares | | Corpus means | |
|---|---|---|---|
| Quadrant | Share | Metric | Value |
| Low-A / Low-E | **0.400** | Mean $H$ | 1.020 |
| High-A only | 0.300 | Mean $A$ | 0.824 |
| High-E only | 0.300 | Mean $E$ | **0.196** |
| Both-high | 0.000 ($<0.001$) | Share $E/H$ | 0.192 |

High/low via 70th percentiles: $\tau_A \approx 0.825$, $\tau_E \approx 0.232$.

## E MUTUAL INFORMATION ON CURATED ROWS

*Why this matters.* Fig. 8 shows that curation reduces epistemic disagreement on edited items.

## F REPRESENTATIVE ANLI EXAMPLES

*What this shows.* Tab. 16 provides concrete premise–hypothesis pairs illustrating adversarial style and label types.

Table 16: Representative ANLI examples. Premises tend to be longer/richer; hypotheses are succinct and often adversarial.

| Premise | Hypothesis | Gold |
|---|---|---|
| "Johnson College Prep is a public four-year charter high school located in the Englewood neighborhood on the south side of Chicago, Illinois, United States. It is part of the Noble Network of Charter Schools. The school is named for African-American businessman John H. Johnson and his wife Eunice Johnson." | The Noble Network of Charter Schools has a school that prepares students for college. | ENT |
| "I would agree. And I also agree that most police officers, of course, are doing a good job and hate this practice also. I talked to an African-American police officer in Springfield, Massachusetts not long ago who raised this question and said that in his opinion one of the biggest solutions is in the training." | Springfield, Massachusetts has few African-American cops. | NEU |
| "How to stop erosion on a river bank: use coir netting that is 700–900 grams per square meter (gsm). Coir netting is made from coconut fibers and is biodegradable. Normal or heavy duty netting should be used when there is a large amount of water flow." | When the water has a large amount of flow use fish netting. | CON |

## G  ABSOLUTE-THRESHOLD UNCERTAINTY QUADRANTS

For sensitivity, we also report the true $2 \times 2$ partition using *absolute* cutoffs shown in Tab. 17.

Table 17: Uncertainty quadrants after refresh with *absolute* cutoffs.

| Quadrant | Share |
|---|---|
| Low-A / Low-E | 0.000 |
| High-A only | 0.149 |
| High-E only | 0.000 |
| Both-high | **0.851** |

Cutoffs: $\tau_A$=0.300, $\tau_E$=0.150. Corpus means after refresh: $A \approx 0.824$, $E \approx 0.196$.

