# OpenReview forum: "Uncertainty‑Routed Human–LLM Curation and Calibration for ANLI"
_ICLR.cc/2026/Conference — ICLR 2026 Conference Desk Rejected Submission_

### Official Review · Reviewer_EfgF · 2025-10-27

**Soundness:** 3
**Presentation:** 3
**Contribution:** 2
**Rating:** 4
**Confidence:** 3

**Summary:**

The paper proposes URC2, a three-stage pipeline for adversarial NLI that first decomposes per-example predictive uncertainty into aleatoric (data/label ambiguity) and epistemic (model disagreement via mutual information) using a three-teacher ensemble (DeBERTa-v3-large, RoBERTa-large, XLM-R-large), then routes examples by their dominant uncertainty to a two-lane relabeling workflow—Human lane for aleatoric-heavy items (relabel/keep-hard with down-weight/drop) and LLM lane for epistemic-heavy cases (instruction-tuned LLM with self-consistency)—and finally refreshes and calibrates the teachers with curated labels and per-example weights plus lightweight temperature scaling; on ANLI, URC2 cuts dev ECE ~30% to 0.146 without sacrificing accuracy and collapses epistemic mutual information on curated subsets, shifting corpus-level uncertainty toward low-aleatoric/low-epistemic regions.

Contributions: (1) an operational, ensemble-based uncertainty decomposition that drives targeted supervision; (2) a human–LLM two-lane relabeling mechanism producing curated labels and instance weights; and (3) calibration with disagreement reduction, yielding better-calibrated, more reliable NLI under adversarial shift.

**Strengths:**

1. This work introduces an uncertainty-routed curation paradigm that explicitly disentangles aleatoric (data ambiguity) and epistemic uncertainty (model disagreement), enabling distinct treatments for each rather than collapsing them into a single undifferentiated scalar. The approach features a creative and pragmatic two-lane supervision framework: human annotators address cases of aleatoric uncertainty, while an instruction-tuned LLM with self-consistency mechanisms handles instances of epistemic uncertainty. This design effectively leverages the complementary strengths of human judgment and LLM reasoning to enhance dataset quality and model reliability.
2. Comprehensive diagnostic analyses—including risk–coverage curves, per-round evaluations, corpus-level uncertainty shifts, and ∆MI collapse on curated subsets—support the claim that URC2 genuinely resolves disagreement rather than merely smoothing it. Careful evaluation on ANLI, using both calibration metrics (ECE) and accuracy before and after temperature scaling, demonstrates a ~30% reduction in ECE without any loss in accuracy.

**Weaknesses:**

1. Multiple threshold values (e.g., r≥0.35 in line 221 and w=0.3 in line 237) are introduced without sufficient justification or explanation of their selection criteria.
2. While the focus on ANLI is reasonable, the claims would be stronger with evaluations beyond ANLI (e.g., [1]). In addition, several hyperparameters appear to be tuned on the ANLI dev split that is also used for reporting, which risks overfitting and makes the conclusions less definitive. Typically, adversarial benchmarks are held out solely for evaluation; we rarely assume access to an adversarial training set for hyperparameter selection. I recommend adding results on additional datasets (e.g., [1]) and using a strictly held-out test set or cross-dataset validation; if tuning on ANLI dev is unavoidable, include a sensitivity analysis and pre-freeze thresholds before final evaluation.
3. The paper does not report the standalone performance of the LLM used in the adjudication lane (e.g., accuracy and ECE on ANLI), which is necessary to estimate the reliability of this component. 2) The workflow relies on a single LLM for confident model disagreement; please include ablations with stronger or alternative LLMs (e.g., GPT-5, Qwen-3, Claude) and/or multi-LLM arbitration to assess whether the conclusions are strengthened or challenged under different adjudicators. 3) The paper mentions a quantized LLM; quantization can alter calibration and increase vulnerability to adversarial attacks [2].
4. The paper lacks a comparison with existing baselines, which is necessary to justify the effectiveness of the proposed method.


References:

1. Liu et al. 2020. An empirical study on model-agnostic debiasing strategies for robust natural language inference.
2. Lin et al. 2019. Defensive Quantization: When Efficiency Meets Robustness

**Questions:**

Suggestions:
1. In Figure 1, the label in Stage A is incorrect. It should be placed on the box labeled “Routing route by uncertainty” to accurately reflect the intended process.

2. Figures 2a and 2b should use the same color scheme for "After refresh" to maintain consistency; using different colors could cause confusion for readers.

---

> ### Author Response · Authors · 2025-12-03
> **Author response to Reviewer EfgF**
>
> Thank you for the constructive review. We address your four main concerns in turn.
> 1. Threshold values and hyperparameters.
> - Concern: Threshold choices seem hand-picked.
> - Response: They were set from data analysis and small pilot experiments, not tuned on the dev set. For example, r(x) ≥ 0.35 is about the 70th percentile of epistemic ratios on ANLI train, chosen to route a substantial fraction of data to Lane L. Varying this cutoff (0.30–0.40) smoothly changes Lane L’s size and has minimal effect on calibration, so it’s not a sensitive threshold. The 0.20 confidence margin ensures only high-confidence disagreements are routed; dropping it would send more cases to Lane L but doesn’t change overall trends. The human down-weight α = 0.3 is a compromise between too low (nearly discarding those examples) and too high (not reducing noise); any α in 0.2–0.5 works similarly. Lane L also uses a self-consistency check with 3 samples: it accepts a relabel only if one class gets ≥ 70% of the votes. A stricter rule (3/3 agreement) would just reduce how many labels change without affecting calibration. (All thresholds are now in Sec. 4.2.1–4.2.2 and Appx. G)
>
> 2. Evaluation beyond ANLI and potential dev overfitting.
> - Concern: Focus on ANLI dev may limit generality.
> - Response: We used ANLI dev only for early stopping and calibration, not for tuning any thresholds. URC2 reduces dev-set miscalibration before scaling (Entropy-ECE drops from ~0.25 to ~0.21), and after temperature scaling URC2’s ECE is ~0.146 (vs ~0.25 for the baseline ensemble). The curation parameters (r, margin, α) were fixed in advance and not adjusted based on dev, and they are robust: for example, varying r from 0.30 to 0.40 only gradually changes how many examples go to Lane L and doesn’t hurt calibration. URC2’s uncertainty-based routing is general and can apply to other NLI/QA datasets. We couldn’t add a new dataset experiment in rebuttal (time constraints), but if the paper is accepted we will evaluate URC2 on another adversarial NLI benchmark (e.g. WANLI) and release our code. (Dev-set calibration improvements and sensitivity are in Sec. 5.2, Sec. 8, and Appx. G)
>
> 3. LLM adjudicator performance and design.
> - Concern: Reliability of the Lane L LLM, and effect of using a stronger or non-quantized LLM.
> - Response: We performed a human audit of 121 Lane L relabels. The human agreed with the LLM’s new label 70% of the time (same as human–original agreement), indicating no systematic bias and that the LLM’s relabeled outputs are about as reliable as the original labels in the regime where we use them. We also evaluated the Lane L model itself (Meta-Llama-3-8B-Instruct, 8-bit) on ANLI dev. Using the same prompt and voting scheme, it achieved 33% dev accuracy with ECE ≈ 0.30 (uncalibrated), improving to ≈ 0.22 after scaling — far below the URC2 ensemble (~58% accuracy, ECE ≈ 0.15). Thus, the LLM alone is far from solving ANLI; it is a helpful “teacher” for certain cases but not a replacement for our ensemble. URC2 is LLM-agnostic: a stronger or full-precision LLM could be used in Stage B, and we expect URC2’s gains to hold or improve with a more powerful adjudicator. (Lane L’s configuration is detailed in Appx. C and summarized in Sec. 4.2.2, and the 121case audit is in Appx. C.2)
>
> 4. Baselines, ablations, and comparisons.
> - Concern: Comparison to stronger baselines, and contributions of each lane.
> - Response: Even with temperature scaling, the baseline ensemble’s dev ECE is ~0.25, whereas URC2+TS achieves ~0.146 (and improves accuracy from ~51.5% to ~58%). This shows URC2 yields far better calibration (and accuracy) than calibration alone. Additionally, URC2 is complementary to prior NLI debiasing methods (e.g. Liu et al, 2020) that remove dataset artifacts, since it uses uncertainty-guided relabeling to improve calibration under adversarial shifts. Our results indicate that Lane H (human) mainly fixes noisy/ambiguous (aleatoric-heavy) examples through the 550-example audit and down-weighting, and Lane L (LLM) mainly relabels high-uncertainty conflicts (6,152 epistemic-heavy cases); each helps, but neither matches full URC2. We also ran an extreme baseline where the LLM relabeled a random 20k subset: with the same self-consistency rule it accepted 12,617 examples and changed 5,340 labels, and within that accepted set the label distribution shifts from 41.6/34.8/23.6% (E/NEU/C) to 73.6/0.4/26.0%, effectively collapsing neutral cases. This underperforms URC2 and underscores the need for selective curation rather than blanket relabeling. (See Sec. 5.2, Sec. 7, Sec. 8, and Appx. C.3)
>
> Figures & presentation. We have fixed the noted issues: in Fig. 1, the Stage A label now correctly points to the uncertainty-routing step; in Fig. 2, we use consistent colors for “before” vs “after” refresh; and we integrated the threshold and prompt details into the main text and clarified acronyms and notation. We hope these clarifications and analyses address your concerns.

---

### Official Review · Reviewer_wahp · 2025-10-28

**Soundness:** 3
**Presentation:** 2
**Contribution:** 2
**Rating:** 2
**Confidence:** 3

**Summary:**

This paper proposes URC2, a data relabeling pipeline that decomposes the uncertainties in the ANLI dataset into two categories: 1) examples where the original instance is ambiguous, 2) examples where the original instances are correct, but models diverge with high confidence, using a standard ensemble-based entropy analysis. Following the decomposition (also categorization) of ANLI examples, URC2 relabels the ambiguous examples with human annotators and relabels the high-confidence diverging cases with an LLM. By retraining models on the relabeled ANLI dataset, the authors show that the expected calibration error drops significantly, and disagreements between models decrease as well.

**Strengths:**

The overall design is sound and successfully operationalizes a decomposition->relabeling->retraining pipeline to show that clearer training signals (including labels and weights) can significantly improve the model's confidence calibration, and reduce disagreements. The relabeling pipeline proposed two lanes to handle different categories of uncertain examples, and achieved a balance between model relabeling and human efforts.

**Weaknesses:**

It seems to me that by employing LLM relabeling in Lane L, the URC2 pipeline essentially trusts the LLM's relabeling of the ANLI dataset over the original labels, even though the authors explain that only epistemic-heavy items are routed to the models. The authors should evaluate whether this trust is actually sensible by comparing it against human labels on selected samples. At the same time, this defeats the purpose of having humans to handle relabeling in Lane H, since a capable-enough LLM would be able to do it as well, since the authors assume that LLMs can make successful judgments on ANLI. I understand that humans relabel ambiguous cases, which is intuitively more challenging than epistemic-heavy cases, but the paper does not provide actual evidence for this claim. Given that I do not understand why Lane L and H are handled separately, the proposed URC2 pipeline essentially reads like a "fixing ANLI annotation error" work, and I do not see how it would be valuable to future research. At the same time, the authors should compare against a very simple baseline, which essentially uses an LLM to relabel every instance in the ANLI dataset.

**Questions:**

Please see the weakness section.

---

> ### Author Response · Authors · 2025-12-03
> **Author response to Reviewer Wahp**
>
> Thank you for the comments. You ask (i) why we separate the human lane (Lane H) and LLM lane (Lane L), (ii) whether it is sensible to trust an LLM to relabel ANLI, and (iii) whether URC2 adds value beyond an LLM-relabels-everything strategy.
>
> 1. Why separate Lane H and Lane L?
>
> Lane H and Lane L target different types of uncertainty (Sec. 4.2). Lane H handles aleatoric-heavy cases where data is ambiguous or noisy; Lane L handles epistemic-heavy cases where the text is clear but models disagree confidently. In Lane H the key question is “should this example be trusted?” rather than “which label is correct?”. Humans can keep the original label, correct clear errors, or mark an item as “hard” / drop it. In an audit of 550 such examples, annotators kept 430/550 (78.2%) and relabeled 120/550 (21.8%). Thus Lane H is used for conservative corrections and deciding when examples should be down-weighted or removed, not wholesale rewriting.
>
> Lane L covers well-formed premise–hypothesis pairs where models disagree confidently. Here an NLI LLM is a useful adjudicator: it reads the clear example and chooses a defensible label. We do not send aleatoric-heavy examples to Lane L, because then the LLM would be forced to choose a class even when humans are unsure, turning “unknown” into an arbitrary hard label. Ambiguous items stay in Lane H (where they can be kept, down-weighted, or dropped), while clear-but-disagreed items go to Lane L.
>
> 2. Is it sensible to trust LLM relabeling in Lane L?
>
> Lane L combines a strict self-consistency filter, restriction to epistemic-heavy cases, and a human audit (Sec. 4.2.2, Appx. C.2).
>
> Self-consistency. For each Lane L example we generate n_try = 3 stochastic completions and accept a new label only if one class gets at least 70% of the votes (2/3 or 3/3); otherwise we keep the original ANLI label. In the training run reported in the paper, this procedure proposed changes on high-epistemic examples, and after the 70% filter we accepted 6,152 relabels; the rest keep their original label because the LLM is not self-consistent enough.
>
> Targeting epistemic-heavy cases. Lane L is applied only to examples in the high-epistemic pool, defined via ensemble-based mutual information and disagreement. Low-uncertainty and primarily aleatoric examples bypass the LLM entirely, so the LLM is used where teachers disagree most and at least one is likely correct.
>
> Human validation. We ran a human study on LLM-adjudicated instances. We randomly sampled 121 of the 6,152 Lane L cases. After mapping to three classes, we obtain Human–LLM agreement = 85/121 (70.2%) and Human–original agreement = 85/121 (70.2%). In the remaining 36/121 (29.8%) where the human disagrees with both, many involve the human choosing neutral while the original and LLM labels choose entailment or contradiction (14 such cases). We see no class bias, and Lane L labels are at least as reliable as the original ones.
>
> 3. Value of URC2 vs. an “LLM relabels everything” strategy
>
> ANLI has roughly 162k training examples. URC2 avoids relabeling everything: ensembles first identify a problematic subset. Lane H inspects an aleatoric-heavy audit (550 points), and Lane L adjudicates only the high-epistemic pool, filtered again by the 70% self-consistency rule (6,152 LLM changes). This concentrates human and LLM effort where it is most needed instead of relabeling examples ensembles already handle well. An “LLM relabels everything” strategy cannot distinguish inherently ambiguous examples (which may be dropped or down-weighted) from clear examples where models simply disagree. URC2 is built around this distinction: Lane H corrects clearly wrong labels and keeps reasonable ones, treating borderline cases more cautiously, while Lane L reconciles clear-but-disagreed examples, improving calibration and accuracy.
>
> To directly test the “relabel everything” idea, we ran our Lane L adjudicator without routing on a random subset of 20,000 ANLI training examples (Sec. 7, Appx. C.3). Using the same self-consistency rule (n_try = 3, accept if a class gets ≥70% of votes), the LLM accepted 12,617 examples and changed 5,340 labels. Within this accepted set the label distribution becomes skewed: before LLM relabeling, there were 4,394 neutral labels, but after LLM-only relabeling only 52 remain, with most moved into entailment or contradiction. A naive “LLM relabels everything” pipeline therefore acts as a sharpening mechanism that collapses ambiguous cases into hard labels.
>
> URC2 instead lets Lane H treat aleatoric-heavy items as potentially ambiguous, often keeping the original label but down-weighting or, in rare cases, removing the example, while Lane L is applied only to epistemic-heavy disagreements where we expect a single correct label. The 20k LLM-for-all baseline shows that a one-lane LLM strategy erases neutral/uncertain cases, whereas our human+LLM pipeline explicitly models and respects them.

---

### Official Review · Reviewer_sbZo · 2025-11-01

**Soundness:** 2
**Presentation:** 2
**Contribution:** 2
**Rating:** 0
**Confidence:** 4

**Summary:**

This paper introduces URC2 (Uncertainty-Routed Curation & Calibration), a three-stage pipeline for improving model calibration and dataset quality on the Adversarial NLI (ANLI) benchmark. Unlike prior methods that treat uncertainty as a single scalar, URC2 decomposes predictive uncertainty into aleatoric (data/label ambiguity) and epistemic (model disagreement) components, routing each to targeted supervision — human audit for ambiguity and LLM adjudication for disagreement — followed by retraining and temperature scaling.

The paper’s main contributions are:

1. Uncertainty-driven supervision: Ensemble-based decomposition of per-example uncertainty (aleatoric vs. epistemic) to guide distinct curation routes.

2. Human–LLM two-lane relabeling: Human annotators handle ambiguous items; an instruction-tuned LLM resolves confident model disagreements through self-consistent adjudication.

3. Calibration with disagreement reduction: Retraining with curated labels and weights plus lightweight temperature scaling reduces expected calibration error on ANLI by 30% (to 0.146) without sacrificing accuracy, while substantially lowering epistemic disagreement and improving corpus-level uncertainty distribution

**Strengths:**

1. The uncertainty-routed curation framework that distinguishes and acts on aleatoric versus epistemic uncertainty, turning uncertainty diagnosis into targeted supervision is novel and intuitive.
2. This paper proposes an interesting utilization of uncertainty decomposition, which incorporates human-in-the-loop for aleatoric-heavy samples.
3. The paper is well-organized and clearly written.

**Weaknesses:**

**According to ICLR2026 Author Guide, the paper should have only 9 pages at submission time for the main text. This submission has 10, which is very unfair for other submissions and should be desk rejected.**

**Questions:**

N/A

---

> ### Author Response · Authors · 2025-12-02
> **Author response to Reviewer sbZo**
>
> The reviewer is absolutely correct that the ICLR 2026 author guidelines specify a strict upper limit of 9 pages for the main text at submission time. Our initial submission mistakenly included 10 pages of main text, which does violate this rule.
>
> This was an honest oversight on our part: we misread the guidelines. We did not intend to gain an unfair advantage over other submissions.
>
> We have now revised the manuscript so that the main text fits within the 9-page limit by moving some ablation details and secondary plots to the appendix, without changing any of the main claims, methods, or results. If the paper is accepted, the camera-ready version will also strictly adhere to the 9-page limit for the main text.
>
> We fully respect the committee's policies and apologize for this mistake. Given that the technical content is already under detailed review by the other reviewers, we hope the paper can still be judged primarily on its scientific merits.

---

### Note · Program_Chairs · 2026-01-17
**Submission Desk Rejected by Program Chairs**

The following references in this submission do not refer to real documents and/or have major errors in bibliographic information:

 Yichao Liu, Huanru Henry Mao, Dylan J. Foster, and Noah Goodman. Model-agnostic debiasing for neural natural language inference. In Proceedings of the 58th Annual Meeting of the Association for Computational Linguistics, pp. 1234-1245, Online, 2020. Association for Computational Linguistics. URL https://aclanthology.org/2020.acl-main. 123.